# A Neural Corpus Indexer for Document Retrieval

**Yujing Wang**[1]    **Yingyan Hou**[1,2,∗]  **Haonan Wang**[1,3,∗]  **Ziming Miao**[1]    **Shibin Wu**[1,2,∗]
**Hao Sun**[1,4,∗]  **Qi Chen**[1]   **Yuqing Xia**[1]    **Chengmin Chi**[1]   **Guoshuai Zhao**[1]    **Zheng Liu**[1]
**Xing Xie**[1]    **Hao Allen Sun**[1]    **Weiwei Deng**[1]    **Qi Zhang**[1]    **Mao Yang**[1]
[1]Microsoft  [2]Tsinghua University  [3]University of Illinois, Urbana Champaign  [4]Peking University
[1] {yujwang, zimiao, cheqi, yuqxia, chec, zhengliu}@microsoft.com
[1] {guzhao, xingx, hasun, dedeng, zhang.qi, maoyang}@microsoft.com
[2] {hyy20, wusb20}@mails.tsinghua.edu.cn
[3] haonan3@illinois.edu   [4] sunhao@stu.pku.edu.cn

## Abstract

Current state-of-the-art document retrieval solutions mainly follow an index-retrieve paradigm, where the index is hard to be directly optimized for the final retrieval target. In this paper, we aim to show that an end-to-end deep neural network unifying training and indexing stages can significantly improve the recall performance of traditional methods. To this end, we propose Neural Corpus Indexer (NCI), a sequence-to-sequence network that generates relevant document identifiers directly for a designated query. To optimize the recall performance of NCI, we invent a prefix-aware weight-adaptive decoder architecture, and leverage tailored techniques including query generation, semantic document identifiers, and consistency-based regularization. Empirical studies demonstrated the superiority of NCI on two commonly used academic benchmarks, achieving +21.4% and +16.8% relative enhancement for Recall@1 on NQ320$k$ dataset and R-Precision on TriviaQA dataset, respectively, compared to the best baseline method.

## 1  Introduction

Document retrieval and ranking are two key stages for a standard web search engine [56, 34]. First, the document retrieval stage retrieves candidate documents relevant to the query, and then, the ranking stage gives a more precise ranking score for each document. The ranking stage is often fulfilled by a deep neural network, taking each pair of query and document as input and predicting their relevance score. Nevertheless, a precise ranking model is very costly, while typically only a hundred or thousand candidates per query are affordable in an online system. As a result, the recall performance of the document retrieval stage is very crucial to the effectiveness of web search engines.

Existing document retrieval methods can be divided into two categories, namely *term-based* and *semantic-based* approaches [22]. Term-based retrieval approaches [9, 59] build an inverted index for the entire web corpus, but they hardly capture document semantics and fail to retrieve similar documents in different wordings. Thus, semantic-based approaches [56, 36] are proposed to alleviate this discrepancy. First, they learn dense representations for both queries and documents through a twin-tower architecture; then Approximate Nearest Neighbor (ANN) search is applied to retrieve relevant documents for the designated query. Despite of their success in real applications, these approaches can not fully leverage the power of deep neural networks for the following reasons. First, a single embedding vector has limited capacity to memorize all semantics in a document, and it performs even worse than term-based methods in the applications that heavily rely on exact match [37]. Second, the model is unable to incorporate deep query-document interactions. Because

---

∗The work was done at Microsoft.

ANN algorithms theoretically require a strong assumption for the Euclidean space, we have to adopt simple functions such as cosine similarity to capture the query-document interactions [20].

Given the above limitations, several research works have explored end-to-end models that directly retrieve relevant candidates without using an explicit index. Gao et al. [20] proposed a Deep Retrieval (DR) framework for item recommendation, which learned a retrievable structure with historical user-item interactions. Nevertheless, it is more challenging to design a universal model for semantic text retrieval, as we need to leverage the power of both pre-trained language models and deep retrieval networks simultaneously. Tay et al. [50] proposed Differentiable Search Index (DSI), a text-to-text model that maps queries directly to relevant docids. To the best of our knowledge, this is the first attempt to propose a differentiable index for semantic search. However, the vanilla transformer decoder in DSI does not fully leverage the hierarchical structures of document identifiers, and the model is pruned to over-fitting with limited training data. Furthermore, Bevilacqua et al. [4] proposed SEAL by leveraging all n-grams in a passage as its identifiers. But for long documents, it is hard to enumerate all possible n-grams. In general, the recall performance of end-to-end document retrieval remains a large room to be improved.

In this paper, we show that the traditional text retrieval frameworks can be fundamentally changed by a unified deep neural network with tailored designs. To this end, we propose a Neural Corpus Indexer (NCI), which supports end-to-end document retrieval by a sequence-to-sequence neural network. The model takes a user query as input, generates the query embedding through the encoder, and outputs the identifiers of relevant documents using the decoder. It can be trained by both ground-truth and augmented query-document pairs. During inference, the top $N$ documents are retrieved via beam search based on the decoder. Designing and training such a model is non-trivial, so we propose several crucial techniques to ensure its effectiveness. First, to get sufficient query-document pairs for training, we leverage a query generation network to obtain possible pairs of queries and documents. Second, we utilize the hierarchical $k$-means algorithm to generate a semantic identifier for each document. Third, we design a prefix-aware weight-adaptive decoder to replace the vanilla one in a sequence-to-sequence architecture. Specifically, the same token will be assigned different embedding vectors at different positions in the identifiers, while another transformer-based adaptive module is applied to the classification weights for token prediction in the context of a certain prefix. This makes the classifiers customized to different prefixes when decoding along the hierarchical tree structure. Besides, a consistency-based regularization loss is taken for training both encoder and decoder networks to mitigate the over-fitting problem.

Our NCI design solves the limitations of traditional index-retrieve pipelines from multiple perspectives. On one hand, a whole neural network model replaces the traditional inverted index or vector search solutions. It can be optimized end-to-end using realistic query-document pairs, which fully captures both term-based and semantic-based features and is adaptive to the changing of workloads. On the other hand, the model is able to capture deep interactions between queries and documents via the encoder-decoder attention, which enlarges the capacity of vector-based representations. Moreover, NCI achieves much better ranking results than ANN-based approaches as it is optimized directly by the final target. Thus, it can be served as an end-to-end retrieval solution while releasing the burden of re-ranking for a long candidate list.

In addition to the superior performance, the invention of Neural Corpus Indexer is also promising from the perspective of system design. As nowadays, ranking and query-answering modules are already implemented by neural networks, NCI finishes the last piece of puzzle for the next-generation information retrieval system based on a unified differentiable model architecture. This reduces the dependency among different sub-modules, while the processes of system deployment and maintenance could be greatly eased.

Our **contributions** are highlighted as follows.

- For the first time, we demonstrate that an end-to-end differentiable document retrieval model can significantly outperform both inverted index and dense retrieval solutions. This finding will inspire research on further steps towards the next-generation search systems, for instance, unifying informational retrieval, ranking, and question answering in a single differentiable framework.

- We design a sequence-to-sequence model, named Neural Corpus Indexer (NCI), which generates relevant document identifiers directly for a specific query. In our experiments, the proposed NCI model improves the state-of-the-art performance of existing methods by a significant margin, achieving +21.4% and +16.8% relative enhancement for Recall@1 on NQ320$k$ dataset and

R-Precision on TriviaQA dataset, respectively. Also, NCI itself achieves a competitive MRR score without using an explicit ranking model.

- We propose a novel decoder architecture, namely *prefix-aware weight-adaptive (PAWA)* decoder, to generate document identifiers. As verified by ablation studies, this invention is very crucial for NCI to achieve an outstanding performance. Moreover, query generation, semantic document identifiers, and consistency-based regularization are all accountable for the superior capability of Neural Corpus Indexer.

## 2   Related work

In this section, we briefly introduce the related works and leave more discussions in Appendix A.

**Sparse retrieval.** Traditional document retrieval methods are based on *Sparse Retrieval*, which is built upon inverted index with term matching metrics such as TF-IDF [45], query likelihood [33] or BM25 [44]. In industry-scale web search, BM25 is a difficult-to-beat baseline owing to its outstanding trade-off between accuracy and efficiency. In recent years, there are some attempts to incorporate the power of neural networks into inverted index. The Standalone Neural Ranking Model (SNRM) [57] learns high-dimensional sparse representations for query and documents, which enables the construction of inverted index for efficient document retrieval. Doc2Query [41] predicts relevant queries to augment the content of each document before building the BM25 index, and DocT5Query [40] improves the performance of query generation by the pre-trained language model T5 [5]. Furthermore, DeepCT [9] calculates context-aware term importance through neural networks to improve the term matching metrics of BM25.

**Dense retrieval.** Another line of research lies in *Dense Retrieval*, which presents query and documents in dense vectors and models their similarities with inner product or cosine similarity. These methods benefit from recent progresses of pre-trained language models, such as BERT [14] and RoBERTa [35] to obtain dense representations for queries and documents. At inference time, efficient Approximate Nearest Neighbor (ANN) search algorithms, such as k-dimensional trees [3], locality-sensitive hashing [10], and graph-based indexes (e.g., HNSW [38], DiskANN [27] and SPANN [7]) can be utilized to retrieve relevant documents within a sublinear time. Besides, Luan et al. [37] analyze the limited capacity of dual encoders, and propose a combination of sparse and dense retrieval methods with multi-vector encoding to achieve better search quality.

**Autoregressive retrieval.** The other way to approach retrieval is utilizing an end-to-end autoregressive model. Firstly, several efforts have been done on entity linking [13, 12, 11], which can be regarded as a special type of retrieval task, *e.g.*, using an entity to ask the posed question. Recently, different from the entity linking task, Tay et al. [50] proposed the DSI (differentiable search index) model to generate relevant document identifiers directly corresponding to the query. Bevilacqua et al. [4] employed the autoregressive model to generate relevant words for a query and utilize the generated string to retrieve relevant documents. Besides, the Deep Retrieval (DR) [20] approach for recommendation is also related to this category, which learns a deep retrievable network with user-item clicks and gets rid of the ANN algorithms based on the Euclidean space assumption.

**Pre-trained language models.** Recently, pre-trained Language Models (LMs), such as BERT [14] and RoBERTa [35], have led to a revolution in web search techniques. The representation vectors for all documents can be calculated and indexed offline. In the online serving stage, it calculates the representation vector for the input query, and applies a crossing layer to calculate the relevance score between each query and document pair. The crossing layer usually adopts simple operators such as cosine similarity or a single feed-forward layer to retain a high efficiency. Gao et al. [16] found that a standard LMs' internal attention structure is not ready-to-use for dense encoders and proposed the Condenser to improve the performance of dense retrieval. Moreover, ANCE [54] leverages hard negatives to improve the effectiveness of contrastive learning, which generates better text representations for the retrieval tasks.

## 3   Neural corpus indexer

The neural corpus indexer (NCI) is a sequence-to-sequence neural network model. The model takes a *query* as input and outputs the most relevant *document identifier (docid)*, which can be trained by a large collection of *<query, docid>* pairs. The documents are encoded into semantic

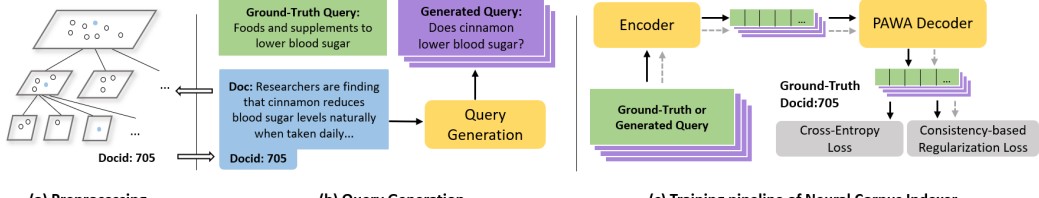

**Figure 1:** Overview of Neural Corpus Indexer (NCI). (a) Preprocessing. Each document is represented by a semantic identifier via hierarchical $k$-means. (b) Query Generation. Queries are generated for each document based on the content. (c) The training pipeline of NCI. The model is trained over augmented *<query, docid>* pairs through a standard transformer encoder and the proposed Prefix-Aware Weight-Adaptive (PAWA) Decoder.

*docids* by the hierarchical $k$-means algorithm [23], which makes similar documents have "close" identifiers in the hierarchical tree. As shown in Figure 1, NCI is composed of three components, including *Query Generation*, *Encoder*, and *Prefix-Aware Weight-Adaptive (PAWA) Decoder*. Query generation is implemented by a sequence-to-sequence transformer model [52] that takes as input the document terms and produces a query as output [41]. The encoder, following the standard transformer architecture, is composed of $M_1$ stacked transformer blocks, which outputs the representation for an input query. For the decoder network, we stack $M_2$ transformer layers. To better align with the hierarchical nature of the semantic identifiers, we propose a weight adaptation mechanism based on another transformer to make the decoder aware of semantic prefixes. At inference time, the top $N$ relevant documents can be easily obtained via beam search. Due to the hierarchical property of semantic identifiers, it is easy to constrain the beam search on the prefix tree so that only valid identifiers will be generated.

## 3.1 Representing document with semantic identifiers

NCI generates document identifiers solely based on the input query without explicit document content, which is difficult when the size of the corpus is very large. Thus, we aim to inject useful priors into the identifiers so that the semantic information of documents can be incorporated in the decoding process. In other words, we hope the documents with similar semantics have close *docids* to facilitate the learning process of NCI. To achieve this, we leverage the hierarchical $k$-means algorithm to encode documents. As shown in Figure 1(a), given a collection of documents to be indexed, all documents are first classified into $k$ clusters by using their representations encoded by BERT [14]. For cluster with more than $c$ documents, the $k$-means algorithm is applied recursively. For each cluster containing $c$ documents or less, each document is assigned a number starting from 0 to at most $c$-1. In this way, we organize all documents into a tree structure $T$ with root $r_0$. Each document is associated with one leaf node with a deterministic routing path $l = \{r_0, r_1, ..., r_m\}$ from the root, where $r_i \in [0, k)$ represents the internal cluster index for level $i$, and $r_m \in [0, c)$ is the leaf node. The semantic identifier for a document is concatenated by the node indices along the path from root to its corresponding leaf node. For documents with similar semantics, the prefixes of their corresponding identifiers are likely to be the same. For simplicity, we set $k = 30$ and $c = 30$ in all experiments, leaving the optimization of these hyper-parameters to future work. The detailed procedure of hierarchical $k$-means will be described in Algorithm 1 in the Appendix B.2.

## 3.2 Query generation

One challenge of generating document identifiers by single query input is how to make the identifiers aware of the document semantics. Since the content of each document is not explicitly known at inference, it must be incorporated into the model parameters during training. To facilitate the training process, we generate a bunch of queries with a query generation module and bind the information of document content through training the sequence-to-sequence model with generated queries and their corresponding document identifiers. In NCI, we utilize two kinds of augmented queries:

**DocT5Query.** We adopt a standard sequence-to-sequence transformer [52] based on the implementation of DocT5Query [1] pre-trained by a large query-document corpus. It takes as input the document terms and produces relevant queries via random sampling. Note that we use random sampling instead of beam search to ensure the diversity of generated queries.

**Document As Query.** Like DSI [50], we also utilize the first 64 terms for each document as queries. Besides, we randomly selected 10 groups of 64 consecutive terms from the whole article as additional queries. This makes the NCI model aware of the semantic meaning of each document.

### 3.3 Prefix-aware weight-adaptive decoder

Given an input query $x$, the probability of generating a document identifier can be written as:

$$p(l|x, \theta) = \prod_{i=1}^{m} p(r_i|x, r_1, r_2, ..., r_{i-1}, \theta_i), \tag{1}$$

where $r_i$ is the $i$-th token in the current identifier; $x$ is the representation output from encoder; $\theta$ denotes the total parameters and $\theta_i$ is the parameter for the $i$-th step.

This probability can be modeled by a transformer-based decoder. For an internal node with level $i$, the probability is calculated by:

$$h_i = \text{TransformerDecoder}(x, h_1, h_2, ..., h_{i-1}; \theta_i), \tag{2}$$

$$p(r_i|x, r_1, r_2, ..., r_{i-1}, \theta_i) = \text{Softmax}(h_i W). \tag{3}$$

Here $h_i$ is the hidden representation for step $i$, which is calculated by a multi-head attention over encoder representation $x$ and token representations of previous decoding steps. The linear classification weight is denoted by $W \in \mathbb{R}^{d \times v}$, $d$ is the hidden dimension size and $v$ is the vocabulary size of identifiers.

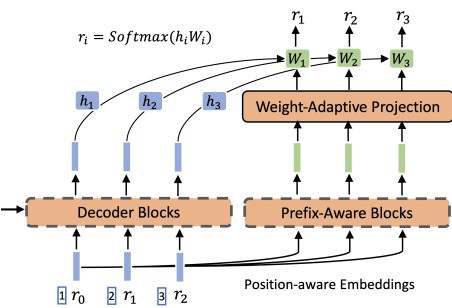

**Figure 2:** Overview of the Prefix-Aware Weight-Adaptive (PAWA) Decoder.

As the encoder and decoder utilize distinct vocabulary spaces, we do not share the embedding space for their tokens. Different from a standard decoding task, the meanings of the same token appearing at different places of the same identifier are different, as they correspond to different clusters in the hierarchical tree structure. For instance, the "$5_2$" and "$5_3$" of the same identifier "$3_1 5_2 5_3$" correspond to different semantic meanings. Moreover, the same token in the same position may have different semantics with different prefixes. For example, in identifiers "$1_1 1_2 5_3$" and "$2_1 4_2 5_3$", the same token "$5_3$" has different semantics in two different identifiers, as they are routed from different prefix paths. These two properties of the hierarchical semantic identifiers motivate us to design the novel **P**refix-**A**ware **W**eight-**A**daptor (PAWA) decoder.

Unlike a standard transformer decoder, the probabilities at different tree levels, such as $p(r_i|x, r_{1..i-1}, \theta_i)$ and $p(r_j|x, r_{1..j-1}, \theta_j)$ where $i \neq j$, do not share parameters with each other. To distinguish different semantic levels, we concatenate the position and token values as input for each decoding step, as shown in the left corner of Figure 2. Specifically, we have "$(1,3)(2,5)(3,5)$" for the semantic identifier "$3_1 5_2 5_3$", while "$(2,5)$" and "$(3,5)$" represent different tokens in the vocabulary space. As the token embedding and linear classification layers share the same weights, the same token value in different positions would correspond to different model parameters. Moreover, to reflect the influence of different prefixes, we expect the linear classification layer to be aware of different prefixes for predicting a specific token. Concretely, instead of using the same projection weight $W$ in the linear classification layer, we employ the prefix-aware adaptive weights for each token classifier, which can be calculated by another transformer decoder,

$$W_{ada}^i = \text{AdaptiveDecoder}(e; r_1, r_2, ..., r_{i-1})W_i \tag{4}$$

where $e$ is the query embedding vector taken as initial input to the transformer decoder; $\{r_t | t \in (1, 2, ..., i-1)\}$ are prefix tokens before the $i$-th position, AdaptiveDecoder stacks $M_3$ transformer decoding layers with dimension $d$, and $W_{ada}^i \in \mathbb{R}^{d \times v}$ is the adapted weight matrix for the corresponding classifier. Finally, the $i$-th token in the given prefix can be predicted by $\text{Softmax}(h_i W_{ada}^i)$.

For instance, to predict the third tokens in the identifiers "$(1,3)(2,1)(3,5)$" and "$(1,2)(2,4)(3,5)$", respectively, the corresponding adaptive weights are derived separately for different prefixes, *i.e.*, "$(1,3)(2,1)$" and "$(1,2)(2,4)$". As we already know the previous tokens for each position in the teacher forcing setting, the prefix-aware adaptive weights can be calculated and trained in parallel in different positions while adding little burden to the entire model.

## 3.4 Training and inference

**Consistency-based regularization.** To alleviate over-fitting, we employ a consistency-based regularization loss for training each decoding step. Given an input query $q$, we denote the decoder representations by two forward passes with independent dropouts before Softmax as $\mathbf{z}_{i,1} = D(r_i|E(q), r_{1,\ldots,i-1}, \theta_i)$ and $\mathbf{z}_{i,2} = D(r_i|E(q), r_{1,\ldots,i-1}, \theta_i)$, respectively, where $E(\cdot)$ denotes the encoder network and $D(\cdot)$ denotes the decoder network. The consistency-based regularization loss tries to distinguish the representations from the same token from those of other tokens, like contrastive learning [8]. The regularization loss of query $q$ for the $i$-th decoding step is defined as,

$$\mathcal{L}_{reg} = -\log \frac{\exp(sim(\mathbf{z}_{i,1}, \mathbf{z}_{i,2})/\tau)}{\sum_{k=1, k \neq 2}^{2Q} \exp(sim((\mathbf{z}_{i,1}, \mathbf{z}_{i,k})/\tau)} \tag{5}$$

where we leverage dot-product for $sim(\cdot)$; $Q$ is the number of queries in the batch, and the temperature parameter is set as $\tau = 1$ in all the experiments.

**Training loss.** Given a set of training examples $\mathcal{D} = \{(q,d)\}$ composed of queries (training queries and augmented queries) and document identifiers, the loss function can be written as follows:

$$\mathcal{L}(\theta) = \sum_{(q,d) \in \mathcal{D}} \left( \log p(d|E(q), \theta) + \alpha \mathcal{L}_{reg} \right), \tag{6}$$

where $p(d|E(q), \theta)$ denotes the probability of generating $d$ with $q$ as the input. The first part is the seq2seq cross-entropy loss with teacher forcing and the second part is the consistency-based regularization loss summed by all decoding steps. The whole process formulates a sequence-to-sequence neural network, which can be optimized end-to-end via gradient descent. The hyper-parameter $\alpha$ denotes a scaling factor of regularization loss, which will be analyzed in Section 4.4.

**Inference via beam search.** In the inference stage, we calculate the query embedding through the encoder network and then perform beam search on the decoder network. Due to the hierarchical nature of *docid*, it is convincing to constrain the beam search decoding process with a prefix tree, which in turn only generates the valid identifiers. The time complexity of beam search is $O(LBF)$, where $L$ is the max length of identifiers (the depth of tree), $B$ is the beam size and $F$ is the max fanout of the tree (30 in our experiments). Given a balanced tree structure built by a corpus with $N$ documents, the average time complexity for beam search is $O(B\log N)$. We leave detailed descriptions of the constrained beam search algorithm in Appendix B.3.

# 4 Experiments

In this section, we empirically verify the performance of NCI and the effectiveness of each component on the document retrieval task, which generates a ranking list of documents in response to a query. In the following, we discuss the datasets and evaluation protocols in Section 4.1, describe the implementation details and baseline methods in Section 4.2, and present empirical results and analyses in Section 4.3 and 4.4, respectively.

## 4.1 Datasets & evaluation metrics

**Datasets.** We conduct our experiments on two popular benchmarks for document retrieval, *i.e.*, the Natural Questions [32] and TriviaQA dataset [29]. Natural Questions (NQ) [32] was introduced by Google in 2019. The version we use is often referred to as NQ320$k$, which consists of 320$k$ query-document pairs, where the documents are gathered from Wikipedia pages and the queries are natural language questions. We use its predetermined training and validation split for evaluation. TriviaQA is a reading comprehension dataset [29], which includes 78$k$ query-document pairs from the Wikipedia domain. Unlike the NQ320$k$ dataset, a query may include multiple answers in TriviaQA.

**Metrics.** We use widely accepted metrics for information retrieval, including Recall@$N$, Mean Reciprocal Rank (MRR) and R-precision. Recall@$N$ measures how often the desired document is hit by the top-$N$ retrieved candidates. MRR calculates the reciprocal of the rank at which the first relevant document is retrieved. R-Precision is the precision after $R$ documents have been retrieved, where $R$ is the number of relevant documents for the query. A high recall means that the ground truth document is contained in the retrieved candidate list, while a high MRR indicates that the corresponding document has already been ranked at the top position without re-ranking.

## 4.2 Implementation details

**Hierarchical semantic identifier.** For semantic identifiers, we apply a hierarchical $k$-means algorithm over the document embeddings obtained through a 12-layers BERT model with pre-trained parameters (provided by HuggingFace [53]). For each hierarchical layer, we employ the default $k$-means algorithm implemented in scikit-learn [42] with $k = 30$. For simplicity, the recursion terminal condition is also set as $c = 30$.

**Query generation.** We leverage the pre-trained model, DocT5Query [40], for query generation. We provide all document contents in NQ320$k$ and TriviaQA datasets to predict augmented query-document pairs. For each document, we generate 15 queries with the first 512 tokens of the document as input and constrain the maximum length of the generated query as 64.

**Training and inference.** The Neural Corpus Indexer is implemented with python 3.6.10, PyTorch 1.8.1 and HuggingFace transformers 3.4.0. We utilize the parameters of the T5 pre-trained model [5] to initialize the encoder and randomly initialize the PAWA decoder. All NCI experiments are based on a learning rate $2 \times 10^{-4}$ for the encoder and $1 \times 10^{-4}$ for the decoder with a batch size 16 per GPU. We set the scaling factor of the consistency-based regularization loss as $\alpha = 0.15$ and the dropout ratio as 0.1. For inference, we apply the partial beam search algorithm to the trained seq2seq model. We set the length penalty and the beam size as 0.8 and 100, respectively. All experiments are based on a cluster of NVIDIA V100 GPUs with 32GB memory. Each job takes 8 GPUs, resulting in a total batch size of 128 ($16 \times 8$).

**Baselines.** We evaluate BM25 on both raw documents and those augmented by DocT5Query by an open-source implementation [2]. The performance of DSI [49] is referred from its original paper as the implementation has not been officially open-sourced. To avoid the difference in data processing, we reproduce SEAL [4] and ANCE [54] by their official implementations. Some baselines for the TriviaQA dataset are directly referred from [58]. We leave the detailed settings in Appendix B.4.

## 4.3 Results

In Table 1 and 2, we compare the empirical results of NCI and corresponding baselines on two benchmarks. We report NCI models based on T5-Base, T5-Large, and ensemble architectures. One can see that even with the T5-Base architecture, NCI outperforms all baselines by a significant margin across four different metrics on both the NQ320$k$ and TriviaQA datasets. Furthermore, an ensemble of five NCI models also brings a large enhancement, because each model is trained individually with a separate semantic identifier generated by a random k-means initialization, making the models complementary to each other. Expect for NCI, SEAL achieves the second best performance. This verifies the superiority of deep text retrieval over traditional sparse and dense retrieval methods. Comparing to SEAL, NCI improves 17.6% for Recall@1, 10.0% for Recall@10, 3.2% for Recall@100, and 14.9% for MRR@100 on the NQ320$k$ dataset. We find that the generated queries have different distributions with the training queries , so we also fine-tune Doc2Query on this dataset for a comparison (denoted by *w/ qg-ft*). Finally, we achieve 72.78% for Recall@1, outperforming SEAL by 21.4%. On the TriviaQA dataset, NCI obtains 7.9% improvement for Recall@5, 5.5% for Recall@20, 6.0% for Recall@100, and 16.8% for R-Precision. As shown in ablation studies, these improvements are owning to the novel designs of PAWA decoder, query generation, semantic identifiers, and consistency-based regularization. We also notice that query generation plays a key role in boosting the retrieval performance. With query generation, the BM25 + DocT5Query method achieves higher performance than the vanilla BM25, especially on the NQ320$k$ dataset. ANCE achieves competitive performance after fine-tuned by the training pairs, but the performance is relatively lower than our NCI model. Moreover, the MRR@100 and R-Precision metrics of NCI are outstanding, indicating that 80% of the queries can be fulfilled without re-ranking on the retrieved document list. This demonstrates the potential of NCI to be served as an end-to-end solution that replaces the entire index-retrieve-rank pipeline in traditional web search engines.

Furthermore, to study the effect of each component, we report ablation results on both NQ320$k$ and TriviaQA datasets in Table 3. In general, all five components are able to improve the performance of document retrieval, which are detailed below.

**w/o DocT5Query.** This configuration removes the training queries generated by *DocT5Query*. According to the results, the query generation model greatly boosts the performance. The result is

**Table 1:** Performance comparison on NQ320$k$ retrieval task. The settings with *qg-ft* refer to query generation by the DocT5Query model fine-tuned on this dataset. Other settings use the original checkpoint of DocT5Query.

| Method | Recall@1 | Recall@10 | Recall@100 | MRR@100 |
|---|---|---|---|---|
| Neural Corpus Indexer (Base) | 65.86 | 85.20 | 92.42 | 73.12 |
| Neural Corpus Indexer (Large) | 66.23 | 85.27 | 92.49 | 73.37 |
| Neural Corpus Indexer (Ensemble) | 70.46 | 89.35 | 94.75 | 77.82 |
| Neural Corpus Indexer *w/ qg-ft* (Base) | 68.91 | 88.48 | 94.48 | 76.17 |
| Neural Corpus Indexer *w/ qg-ft* (Large) | 68.65 | 88.45 | 94.53 | 76.10 |
| **Neural Corpus Indexer *w/ qg-ft* (Ensemble)** | **72.78** | **91.76** | **96.22** | **80.12** |
| DSI (Base) [50] | 27.40 | 56.60 | – | – |
| DSI (Large) [50] | 35.60 | 62.60 | – | – |
| DSI (XXL) [50] | 40.40 | 70.30 | – | – |
| SEAL (Base) [4] | 56.98 | 79.97 | 91.39 | 65.48 |
| SEAL (Large) [4] | **59.93** | **81.24** | 90.93 | **67.70** |
| ANCE (FirstP) [54] | 51.33 | 80.33 | **91.78** | 61.71 |
| ANCE (MaxP) [54] | 52.63 | 80.38 | 91.31 | 62.84 |
| BERT + BruteForce [15] | 28.65 | 53.42 | 73.16 | 36.60 |
| BERT + ANN (Faiss) [28] | 27.92 | 53.63 | 73.01 | 37.08 |
| BM25 + DocT5Query [40] | 35.43 | 61.83 | 76.92 | 44.47 |
| BM25 [44] | 15.11 | 32.48 | 50.54 | 21.07 |

**Table 2:** Performance comparison on TriviaQA retrieval task. The results annotated by * are taken from [58].

| Method | Recall@5 | Recall@20 | Recall@100 | R-Precision |
|---|---|---|---|---|
| Neural Corpus Indexer (Base) | 90.49 | 94.45 | 96.94 | 73.90 |
| Neural Corpus Indexer (Large) | 91.73 | 95.17 | 97.44 | 74.94 |
| **Neural Corpus Indexer (Ensemble)** | **94.60** | **96.89** | **98.20** | **80.84** |
| SEAL (Base) [4] | 86.3 | 90.5 | 91.5 | 68.1 |
| SEAL (Large) [4] | **87.7** | **91.8** | **92.6** | **69.2** |
| AR2-G*[58] | 78.2 | 84.4 | 87.9 | – |
| coCondenser*[18] | 76.8 | 83.2 | 87.3 | – |
| Condenser*[17] | – | 81.9 | 86.2 | – |
| Individual Top-k*[46] | 76.8 | 83.1 | 87.0 | |
| Joint Top-k*[46] | 74.1 | 81.3 | 86.3 | |
| RDR*[55] | - | 82.5 | 87.3 | - |
| ANCE*[54] | - | 80.3 | 85.3 | - |
| DPR*[30] | - | 79.3 | 84.9 | - |
| GAR*[39] | 73.1 | 80.4 | 85.7 | - |
| BM25 + DocT5Query [40] | 59.71 | 72.06 | 82.71 | 39.66 |
| BM25 [44] | 56.91 | 69.45 | 80.24 | 37.29 |

aligned with our expectation because training with augmented queries allows the NCI model to better understand the semantic meanings of each document.

**w/o document as query.** Similar to DSI [49], using the document contents as queries also makes the model aware of the semantics of documents.

**w/o PAWA decoder.** This configuration removes the adaptive decoder layer in Equation (4) and leverages shared weights with token embedding for the linear classification layer. We notice that the prefix-aware weight-adaptive decoder has a noticeable influence on the performance, which indicates that, instead of borrowing the vanilla transformer decoder, it is necessary to design a tailored decoder architecture for the task of semantic identifier generation.

**Table 3:** Ablation Study on NQ320$k$ and TriviaQA retrieval task.

| Method | NQ320$k$ | | | | TriviaQA | | | |
|---|---|---|---|---|---|---|---|---|
| | Recall@1 | Recall@10 | Recall@100 | MRR@100 | Recall@5 | Recall@20 | Recall@100 | R-Precision |
| **Neural Corpus Indexer (Base)** | **65.86** | **85.20** | **92.42** | **73.12** | **90.49** | **94.45** | **96.94** | **73.90** |
| w/o DocT5Query | 60.23 | 80.20 | 90.92 | 67.89 | 84.56 | 90.94 | 95.32 | 63.50 |
| w/o document as query | 62.49 | 81.21 | 88.85 | 69.41 | 85.34 | 91.10 | 94.66 | 67.48 |
| w/o PAWA decoder | 63.36 | 83.06 | 91.47 | 70.56 | 88.75 | 93.56 | 96.18 | 71.81 |
| w/o semantic id | 62.75 | 83.88 | 91.01 | 70.43 | 88.91 | 93.07 | 95.80 | 72.57 |
| w/o regularization | 65.07 | 82.91 | 90.65 | 71.80 | 89.01 | 93.63 | 96.16 | 71.59 |
| w/o constrained beam search | 65.65 | 84.89 | 92.23 | 72.79 | 89.58 | 93.97 | 96.61 | 72.51 |

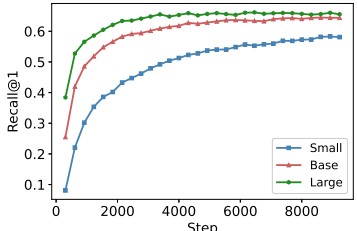 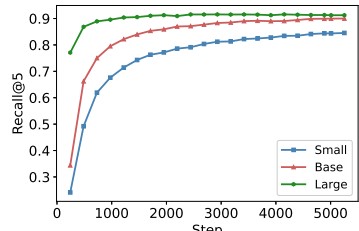

**Figure 3:** Learning curves of NCI with different model capacities. **Left:** NQ320$k$; **Right:** TriviaQA.

**w/o semantic id.** This configuration replaces the semantic identifier of each document to a randomly generated one. We find a relative drop in the model performance on all four metrics, demonstrating that the semantic identifiers derived by the hierarchical $k$-means have injected useful priors. We conjecture that the performance enhancement would be more significant on a larger document corpus.

**w/o regularization.** There is a performance drop on all four metrics without using consistency-based regularization loss. The reason is that the decoder network is prone to over-fitting. By making the prediction results of two augmented queries consistent, the decoder will become more generalizable and resistant to over-fitting.

**w/o constrained beam search.** This configuration disables the validating constraint in beam search. In other words, the decoder network does not have a tree-based prior structure. Instead, all tokens in the vocabulary can be generated in each decoding step. We observe a performance drop on four evaluation metrics. This indicates that it is difficult to remember all information of valid identifiers in the network, and an explicit prior could be helpful for improving the quality of beam search.

**Table 4:** NCI with different number of layers in PAWA adapter. **Left:** NQ320$k$; **Right:** TriviaQA.

| Setting | Recall@1 | Recall@10 | Recall@100 | MRR@100 |
|---|---|---|---|---|
| #layer = 0 | 63.36 | 83.06 | 91.47 | 70.56 |
| #layer = 1 | 64.85 | 84.71 | 91.49 | 71.42 |
| #layer = 2 | 65.40 | 85.12 | **92.82** | 72.83 |
| #layer = 4 | **65.86** | **85.20** | 92.42 | **73.12** |
| #layer = 6 | 65.07 | 83.91 | 91.65 | 71.80 |
| #layer = 8 | 63.60 | 83.11 | 91.78 | 71.22 |

| Setting | Recall@5 | Recall@20 | Recall@100 | R-Precision |
|---|---|---|---|---|
| #layer = 0 | 88.75 | 93.56 | 96.18 | 71.81 |
| #layer = 1 | 89.16 | 93.90 | 96.58 | 70.77 |
| #layer = 2 | 89.89 | 94.35 | 96.86 | 72.89 |
| #layer = 4 | **90.49** | **94.45** | **96.94** | **73.90** |
| #layer = 6 | 89.76 | 94.31 | 96.76 | 73.32 |
| #layer = 8 | 87.90 | 93.30 | 96.20 | 70.75 |

## 4.4 Analysis

**Model capacity.** Figure 3 compares the learning curves of NCI with different model capacities, which are identical to the small, base, and large settings of ordinary T5 [43]. We observe that with the increase of model size, NCI convergences more quickly with fewer epochs. At convergence, the small model achieves a relatively lower recall. Instead, both the base and large models achieve similar results after sufficient training epochs, and the large model will be slightly higher. This implies that the model capacity has a critical impact on the retrieval performance, and the capacity of base model seems to be enough to memorize all documents in NQ320$k$ and TriviaQA datasets. The large model can be used when the computation capacity is sufficient. For a larger corpus, one may need to increase the model size to obtain satisfactory performance.

**Layer number of PAWA adapter.** We study the influence of the number of transformer layers in the PAWA adapter and choose the layer number from {0,1,2,4,6,8}. The results are summarized in Table 4. We notice that with the increase of layer number, *i.e.* from 0 to 4, the overall performance is consistently improved on four metrics. But when the number of layers achieves 6, the performance decreases. When continuing to increase the number of layers to 8, the performance drops significantly. We attribute that to the overfitting issue caused by a large PAWA decoder. Therefore, we adopt the PAWA decoder with a 4-layers adapter in NCI.

**Retrieved documents and their semantics identifiers.** To verify the effectiveness of retrieval as well as the semantic identifiers learned by the hierarchical $k$-means, we analyze the retrieval results of NCI for some exemplar queries. To illustrate, we select four queries denoted by *A-1*, *A-2*, *B-1* and *B-2*, where two queries inside the same group are semantically similar, and the queries in different groups correspond to distinct topics. In Figure 4, we show the probabilities of retrieved documents

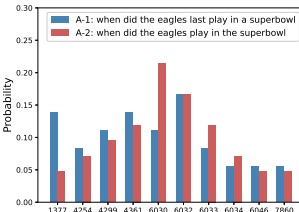 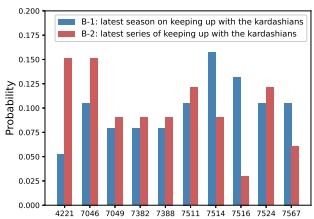 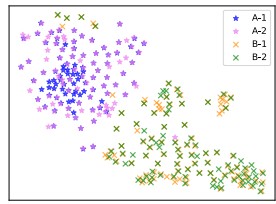

**Figure 4:** Analyses of retrieved documents with semantic identifiers. **Left:** The probabilities of retrieved documents for Query Group A; **Middle:** The probabilities for Query Group B; **Right:** The t-SNE visualization of BERT-based document embeddings.

for each query in group *A* and *B*, respectively. The digits along x-axis denote the four-bit prefixes for the semantic identifiers of retrieved documents, and the y-axis stands for their probabilities. We notice that similar queries result in close document distributions, while dissimilar queries in different groups result in un-overlapped document collections. In addition, the documents retrieved by the same group of queries have close prefixes for the identifiers, *e.g.*, 6030, 6032, 6033, 6034 in group *A* and 7511, 7514, 7516 in group *B*. Also, we visualize the BERT-based document embeddings by t-SNE [51] in Figure 4, in which each color represents the corresponding documents for a specific query. As shown in the figure, these documents naturally form two clusters with respect to different query groups. Thus, we conclude that the semantic document identifiers generated by the hierarchical $k$-means algorithm have positive effects on the retrieval performance.

**Efficiency Analysis.** We use an NVIDIA V100-32G GPU to analyze the efficiency of NCI. As the inference speed is influenced by both model capacity and beam size, we report the latency and throughput measures for multiple settings in Table 5. As NCI is an end-to-end retrieval method and achieves competitive performance without re-ranking, the latency and throughput are already affordable for some near-real-time applications. The latency of NCI is on par with DSI and SEAL using the same model size and beam

**Table 5:** Efficiency analysis

| Model size | Beam size | Latency (ms) | Throughput (queries / s) |
|---|---|---|---|
| Small | 10 | 78.46 | 58.48 |
| Base | 10 | 115.17 | 52.55 |
| Large | 10 | 188.60 | 43.39 |
| Small | 100 | 216.01 | 6.12 |
| Base | 100 | 269.31 | 5.62 |
| Large | 100 | 356.07 | 4.75 |

size, because all of them conduct beam search based on transformer decoders. BM25 is very efficient (<100ms per query on CPU using an open-source implementation [2]), but the recall metrics are much lower. Furthermore, we can leverage other techniques to improve the efficiency of NCI, which will be discussed in the later section.

## 5 Limitation & Future Works

Despite the significant breakthrough, the current implementation of NCI still suffers from several limitations before deployment in a large-scale search system. Firstly, it requires a much larger model capacity for extending NCI to the web scale. Secondly, the inference speed needs to be improved to serve online queries in real time. Thirdly, it is difficult to update the model-based index when new documents are added to the system. In future works, we may tackle these problems from four aspects. (1) The architecture of sparsely-gated Mixture of Expert (MoE) [47] can be employed to enhance the model capacity. (2) Documents can be grouped into semantic clusters, and NCI can be used to retrieve relevant cluster identifiers. In this way, all documents in relevant clusters can be retrieved efficiently. (3) Model compression techniques, like weight quantization [26] and knowledge distillation [24], can be further taken to speed up inference. (4) We plan to explore a hybrid solution by building another index that serves new documents through traditional indexing algorithms.

## 6 Conclusion

In this work, we introduce a novel document retrieval paradigm that unifies the training and indexing stages by an end-to-end deep neural network. The proposed Neural Corpus Indexer (NCI) directly retrieves the identifiers of relevant documents for an input query, which can be optimized end-to-end using augmented query-document pairs. To optimize the recall and ranking performance, we invent a tailored prefix-aware weight-adaptive decoder. Empirically, we evaluate NCI on NQ320$k$ and TriviaQA datasets, demonstrating its outstanding performance over state-of-the-art solutions.

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
