# A   Related work

Traditional web search techniques follow a two-stages paradigm including *document retrieval* and *document ranking*. The first stage aims to select a collection of documents relevant to a given query, which requires an ingenious trade-off between efficiency and recall. Then, the document ranking stage takes more advanced features and deeper models to calculate a fine-grained ranking score for each query and document pair. In the following, we first discuss related works for document retrieval and ranking respectively. Afterwards, we introduce recent works that incorporate pre-trained language models into these two stages. At last, the attempts on end-to-end retrieval will be discussed.

## A.1   Document retrieval

Traditional document retrieval methods are based on *Sparse Retrieval*, which is built upon inverted index with term matching metrics such as TF-IDF [45], query likelihood [33] or BM25 [44]. In industry-scale web search, BM25 is a difficult-to-beat baseline owing to its outstanding trade-off between accuracy and efficiency. In recent years, there are some attempts to incorporate the power of neural networks into inverted index. The Standalone Neural Ranking Model (SNRM) [57] learns high-dimensional sparse representations for queries and documents, which enables the construction of inverted index for efficient document retrieval. Doc2Query [41] predicts relevant queries to augment the content of each document before building the BM25 index, and DocT5Query [40] improves the performance of query generation by the pre-trained language model T5 [5]. Furthermore, DeepCT [9] calculates context-aware term importance through neural networks to improve the term matching metrics of BM25.

Another line of research lies in *Dense Retrieval*, which presents query and documents in dense vectors and models their similarities with inner product or cosine similarity. These methods benefit from recent progresses of pre-trained language models, such as BERT [14] and RoBERTa [35] to obtain dense representations for queries and documents. At inference time, efficient Approximate Nearest Neighbor (ANN) search algorithms, such as k-dimensional trees [3], locality-sensitive hashing [10], and graph-based indexes (e.g., HNSW [38], DiskANN [27] and SPANN [7]) can be utilized to retrieve relevant documents within a sublinear time. Besides, Luan et al. [37] analyze the limited capacity of dual encoders, and propose a combination of sparse and dense retrieval methods with multi-vector encoding to achieve better search quality.

## A.2   Document ranking

Document ranking has been extensively studied in recent years and experienced a huge improvement with the booming of deep neural networks. Neural network-based document ranking models mainly fall into two categories. *Representation-based models* like DSSM (Deep Structured Semantic Model) [25] and CDSSM (a convolution-based variant of DSSM) [48] represent query and document in a shared semantic space and model their semantic similarity through a neural network. In contrast, *Interaction-based models* first build interactions between query and document terms, and then utilizes neural networks to learn hierarchical interaction patterns. For example, DRMM (Deep Relevance Matching Model) [21] extracts interactive features by matching histograms and utilizing a feed forward network with term-gating mechanism to calculate the relevance score of a query-document pair.

## A.3   Pre-trained language models

Recently, Pre-trained Language Models (PLMs) like BERT [14] have led to a revolution of web search techniques. The vanilla BERT model utilizes a single-tower architecture that concatenates query and document tokens as a whole input to the relevance model. Despite of its superior performance, the high computational cost hinders its application to industrial-scale web search systems. TwinBERT [36] tackles this problem by exploiting a Siamese architecture, where queries and documents are first modeled by two BERT encoders separately, and then an efficient crossing layer is adopted for relevance calculation. The representation vectors for all documents can be calculated and indexed offline. In the online serving stage, it calculates the representation vector for the input query and applies a crossing layer to calculate the relevance score between each query and document. The

crossing layer usually adopts simple similarity functions such as dot product or a single feed-forward layer to achieve a high efficiency.

Moreover, Chang et al. [6] argue that the Masked Language Model (MLM) loss designed for BERT pre-training is not naturally fitted to embedding-based retrieval tasks. Instead, they propose three paragraph-level pre-training tasks, *i.e.*, Inverse Cloze Task (ICT), Body First Selection (BFS), and Wiki Link Prediction (WLP), which demonstrate promising results in text retrieval experiments. Gao et al. [16] find that a standard LMs' internal attention structure is not ready-to-use for dense encoders. Thus, they propose a novel architecture named Condenser to improve the performance of dense retrieval. ANCE (Approximate nearest neighbor Negative Contrastive Estimation) [54] leverages hard negatives to improve the effectiveness of contrastive learning, which generates better text representations for the retrieval task.

### A.4 End-to-end retrieval

The deficiency of index-retrieve paradigm lies in that the two stages of document retrieval and re-ranking are optimized separately. Especially, the document retrieval procedure is often sub-optimal and hinders the performance of the entire system. Thus, there are some recent attempts to achieve end-to-end retrieval as a one-stage solution. ColBERT [31] introduces a contextualized late interaction architecture, which independently encodes query and document through BERT, and performs cross-term interaction based on the contextualized representations of query and document terms. ColBERT supports end-to-end retrieval directly from a large document collection by leveraging vector-similarity indexes in the pruned interaction layer. It can be viewed as a compromise between single-tower and twin-tower BERT architectures which maintains an effective trade-off between accuracy and latency. Moreover, the Contextualized Inverted List (COIL) [19] exacts lexical patterns from exact matching pairs through contextualized language representations. At search time, we build representation vectors for query tokens and perform contextualized exact match to retrieve relevant documents based on inverted index.

Although ColBERT and COIL have shown promising results in end-to-end retrieval tasks without re-ranking, their performance is still not obviously better (if not worse) than a common practice of "BM25 indexer + BERT re-ranker", and their efficiency is also not good enough for an industrial web search engine. Therefore, we resort to a new indexing paradigm to break the bottleneck. We believe the neural corpus indexer proposed in this paper is a crucial break-through, opening up new opportunities to optimize the performance of web-scale document retrieval. Moreover, there are a few attempts that try to build a model-based search index by directly predicting document identifiers. Tay et al. [50] proposed the DSI (differentiable search index) model based on an encoder-decoder architecture to generate relevant docids. However, its decoder architecture remains the same as T5, which is unsuitable to generate semantic ids derived by hierarchical $k$-means. SEAL [4] uses all n-grams in a passage as its possible identifiers and build a FM-Index to retrieve documents; but it is hard to enumerate all n-grams for retrieving relevant documents. In addition, our work is related to Deep Retrieval [20] for the recommendation task, which learns a deep retrievable network with user-item clicks without resorting to ANN algorithms constrained by the Euclidean space assumption.

## B Reproducibility

We provide our code for reproduction in the supplementary material. We will release it to public shortly.

### B.1 Dataset processing

We conduct experiments on NQ320$k$ and TriviaQA datasets. For NQ320$k$ dataset, the queries are natural language questions and the documents are Wikipedia articles in HTML format. During dataset processing, we first filter out useless HTML tag tokens, and extract title, abstract and content strings of each Wikipedia article using regular expression. The experiments are also conducted on TriviaQA dataset. For TriviaQA dataset, it includes 78$k$ query-document pairs from the Wikipedia domain, which are processed almost the same as NQ320$k$. Then, we detect duplicated articles based on the title of each article. After that, we concatenate the title, abstract and content strings of each Wikipedia article, and apply a 12 layers pre-trained BERT model on it to generate document embeddings.

Finally, hierarchical $k$-means is applied on the article embeddings to produce semantic identifiers for each article.

## B.2 Hierarchical $k$-means for semantic identifier

The pseudo code of hierarchical $k$-means is detailed in in Algorithm 1.

---

**Algorithm 1:** Hierarchical $k$-means.

---

**Input:**
  Document embedding $X_{1:N}$
  Number of clusters $k$
  Recursion terminal condition $c$
**Output:**
  Hierarchical semantic identifier $L_{1:N}$
**Function:**
  **GenerateSemanticIdentifier**($X_{1:N}$)

  $C_{1:k} \leftarrow KMeansCluster(X_{1:N}, k)$
  $L \leftarrow \varnothing$
  **for** i $\in [0, k-1]$ **do**
    $L_{current} \leftarrow [i] * |C_{i+1}|$
    **if** $|C_{i+1}| > c$ **then**
      $L_{rest} \leftarrow$ GenerateSemanticIdentifier($C_{i+1}$)
    **else**
      $L_{rest} \leftarrow [0, ..., |C_{i+1}| - 1]$
    **end if**
    $L_{cluster} \leftarrow$ ConcatString($L_{current}, L_{rest}$)
    $L \leftarrow L$.Append($L_{cluster}$)
  **end for**
  $L \leftarrow$ reorderToOriginal($L, X_{1:N}, C_{1:k}$)
  **return** $L$

---

## B.3 Constrained beam search

The pseudo code of constrained beam search is detailed in Algorithm 2.

## B.4 Baselines

We describe the baseline methods in this section. For most of them, we use their official open-source implementations.

- **BM25.** BM25 is currently the mainstream algorithm for calculating the similarity score between query and document in information retrieval [44]. We calculate BM25 between an original query $Q$ and a document $d$ which derived from a sum of contributions from each query term $q_i$ as,

$$\text{Score}(Q, d) = \sum_{i=1}^{t} w_i * R(q_i, d) \tag{7}$$

where $w_i$ denotes the weight of $q_i$, and $R(q_i, d)$ is the correlation between $q_i$ and $d$. We use the open-source implementation from Rank-BM25 [2].

- **BM25 + DocT5Query.** The docT5Query model [40] generates questions that related to a document. These predicted queries are then appended to the original documents, which are then indexed. Note that we use the same predicted queries in our query generation module. Queries are issued against the index as "bag of words" queries, using BM25 for evaluation. We use the open-source code for DocT5Query[3], and the generated queries keep the same with NCI (our model) to have a fair comparison.

---

[2]https://github.com/dorianbrown/rank_bm25
[3]https://github.com/castorini/docTTTTTquery

**Algorithm 2:** Constrained Beam Search.

**Input:**
    Query embedding $x$
    Beam search size $k$
    Max beam length $n$
    Prefix tree $T$ with root $r_0$, containing all valid identifiers
    Log probability function $f(r_i) = log(p(r_i|x, r_{i-1}, ..., r_0))$

**Output:**
    $k$ documents with the highest probabilities

**Function:**
    **prefix**=$\{r_0\}$
    **ResultIds** $\leftarrow \varnothing$
    $\mathbf{B_0} \leftarrow \{\langle\mathbf{prefix}, \text{EOS}\rangle\}$
    **for** $i \in [1, n]$ **do**
      **for** $\langle\mathbf{prefix}, sum\_log\_prob\rangle \in \mathbf{B_{i-1}}$ **do**
        **if** **prefix**.last().isLeaf() **then**
          $doc\_id$ = **prefix**.toString()
          **ResultIds**.add($\langle doc\_id, sum\_log\_prob/len(\mathbf{prefix})\rangle$)
        **else**
          **for** $r_i \in$ **prefix**.last().child() **do**
            **new_prefix** = **prefix**.copy().append($r_i$)
            $\mathbf{B_i}$.add($\langle\mathbf{new\_prefix}, sum\_log\_prob + f(r_i)\rangle$)
          **end for**
        **end if**
      **end for**
      $\mathbf{B_i} \leftarrow \mathbf{B_i}$.rank_by_prob().top($k$)
    **end for**
    **return ResultIds**.rank_by_prob().top($k$)

- **BERT + ANN (Faiss).** We use the Flat Index method with the query and document representations obtained by CoCondenser[4][16] which is pretrained on Wikipedia and then finetuned over NQ dataset. For the Flat Index method, we use the version implemented by Faiss[5].

- **BERT + BruteForce.** In this baseline, we use the CoCondenser [16], pretrained on Wikipedia and then finetuned over NQ dataset, to encode queries and documents separately. Then, the Cosine Similarity is computed for each query and document pair. After that, for each query, the documents with the largest Cosine Similarity score are retrieved.

- **ANCE (MaxP & FirstP).** ANCE, a training mechanism, that constructs negatives from an Approximate Nearest Neighbor (ANN) index of the corpus [54]. For BERT FirstP, we concatenate the title and content of each document by a [SEP] token. For BERT MaxP, we only use the content of each document. We use the open-source implementation[6].

- **SEAL (BART-Large).** We reproduce SEAL based on the open-sourced implementation[7].

- **DSI.** The DSI model learns a text-to-text model that maps string queries directly to relevant docids [50]. We report the performance of DSI (T5-Base), DSI (T5-Large) and DSI (T5-XXL) from its original paper as the implementation has not been open-sourced.

## C  More Experimental Results

We study the influence of regularization strength and choose the regularization hyper-parameter $\alpha$ from $\{0, 0.1, 0.15, 0.2, 0.3\}$. Table 6 summaries the results with different regularization hyper-parameter $\alpha$ settings. At convergence, the hyper-parameter $\alpha = 0.15$ generally achieves better performance. Therefore, we set the default value as $\alpha = 0.15$ in NCI.

---

[4]https://github.com/luyug/Condenser
[5]https://github.com/facebookresearch/faiss
[6]https://github.com/microsoft/ANCE
[7]https://github.com/facebookresearch/SEAL

**Table 6:** Different regularization hyper-parameter $\alpha$ in loss function. **Left:** NQ320$k$; **Right:** TriviaQA.

| Setting | Recall@1 | Recall@10 | Recall@100 | MRR@100 |
|---|---|---|---|---|
| $\alpha = 0$ | 65.07 | 82.91 | 90.65 | 71.80 |
| $\alpha = 0.1$ | 65.51 | **85.28** | **92.52** | 72.76 |
| $\alpha = 0.15$ | **65.86** | 85.20 | 92.42 | **73.12** |
| $\alpha = 0.2$ | 65.55 | 84.48 | 92.61 | 72.63 |
| $\alpha = 0.3$ | 65.44 | 85.21 | 92.45 | 72.83 |

| Setting | Recall@5 | Recall@20 | Recall@100 | R-Precision |
|---|---|---|---|---|
| $\alpha = 0$ | 89.01 | 93.63 | 96.16 | 71.59 |
| $\alpha = 0.1$ | 90.14 | 94.15 | 96.96 | 72.78 |
| $\alpha = 0.15$ | **90.49** | **94.45** | 96.94 | **73.90** |
| $\alpha = 0.2$ | 90.44 | 94.41 | **96.97** | 73.22 |
| $\alpha = 0.3$ | 90.02 | 94.09 | 96.79 | 73.53 |

# D  Miscellaneous

**Social Impacts.**  This work aims at introducing a new learning paradigm that can unify the learning and indexing stages with an end-to-end deep neural network. Besides, our work has the potential to inspire more attempts at unifying the retrieval and re-ranking task with an end-to-end framework, which might have positive social impacts. We do not foresee any form of negative social impact induced by our work.

**Privacy Information in Data.**  We use the NQ dataset privided by the work [32]. The dataset only includes questions, rendered Wikipedia pages, tokenized representations of each page, and the annotations added by our annotators. No privacy information is included. For the TriviaQA [29], which is a reading comprehension dataset, it includes 78$k$ query-document pairs from the Wikipedia domain. Again, no privacy information is included.