# OpenReview forum: "A Neural Corpus Indexer for Document Retrieval"
_NeurIPS.cc/2022/Conference — NeurIPS 2022 Accept_

### Official Review · Reviewer_xW4j · 2022-07-11

**Rating:** 8
**Confidence:** 3
**Soundness:** 4 excellent
**Presentation:** 4 excellent
**Contribution:** 3 good

**Summary:**


This paper proposes a new framework for neural IR: given query, directly predict a document ID. The document IDs are obtained by hierarchical clustering of documents beforehand. This is a novel formulation of the problem, and is very distinct from current two-stage methods that have a high-recall sparse retrieval stage, followed by a high-precision neural reranker, or approximate nearest neighbor methods that encode both documents and queries as vectors.

**Questions:**

- Can you provide some timing analysis for the baselines (e.g. in footnote), so Table 4 has some reference of comparison?

**Limitations:**

- The limitation section is reasonable. I think the index-update aspect may be one of the most important near-term challenges.

**Strengths And Weaknesses:**


Strength:
- The formulation is unique and intriguing. I really like this idea! Despite some limitations, the results are very promising and I imagine many follow-up work may come from this paper.

Weakness:
- I think the method rests on one large assumption, that documents can be pre-clustered into some fixed set of IDs. All queries are being trained with the same fixed set of document IDs. This is a somewhat unconventional way to think about IR, as one usually does not assume document cluster structure, and ad hoc queries may flexibly retrieve any subset. I think some experiments that try different definitions of document IDs (e.g. random) would shed light on the assumption of having a strong document clustering prior.

---

> ### Author Response · Authors · 2022-08-02
> **Appreciate your recognition of the work.**
>
> We respond to your major comments for a discussion.
>
> 1.Some experiments that try different definitions of document
> (e.g., random) would shed light on the assumption of having a strong document clustering prior.
> * In the ablation experiments (table 2), we have the “*w/o semantic id*” setting, which does not use semantic IDs generated by pre-clustered priors. Instead, we assign a unique ID randomly to each document. As shown in Table 2, Recall@1 drops from 88.72 to 87.22 for the NCI model trained by augmented queries. Also, it drops from 53.63 to 50.78 in the setting without query generation (see the result table in our general response). Therefore, NCI already achieves good performance using random identifiers, while the pre-clustering prior brings additional benefits to the model. We appreciate your advice and will analyze other definitions of document IDs for more insights in the next step.
>
> 2.Can you provide some timing analysis for the baselines (e.g. in footnote), so Table 4 has some reference of comparison?
> * Thank you for the suggestion. The latency of NCI is on par with DSI and SEAL with the same model size and beam size as all of them conduct beam search based on transformer decoders. We also test the latency of BM25 for reference. We leverage an open-source implementation of BM25 [1] and test it on Intel(R) Xeon(R) CPU E5-2690 v4@2.60GHz. The latency is less than 100ms per query. NCI/DSI/SEAL can also achieve near real-time serving on GPU, but it hardly to be served on CPU. We have other proposals to improve the efficiency of NCI, which is discussed in the section of future work.
>
> 3.The index-update aspect may be one of the most important near-term challenges.
> * We concur with you. We have an initial solution: take each leaf node in the tree as a virtual semantic cluster and assign a specific cluster to each new document. Then, in the inference stage, the new document is retrieved along with the generated cluster ID. We are working in this direction and plan to address this problem in the next version.
>
> [1] https://github.com/castorini/anserini

---

### Official Review · Reviewer_RUss · 2022-07-12

**Rating:** 4
**Confidence:** 5
**Soundness:** 3 good
**Presentation:** 2 fair
**Contribution:** 2 fair

**Summary:**

The paper focuses on a promising direction to learn retrieval via generation, where the model will generate the relevant document IDs for given queries. It resembles the previous Differentiable Search Index work with several minor improvements.


**Questions:**

NA

**Strengths And Weaknesses:**

Strengths:
They propose several techniques to improve the model such as prefix-aware weight-adaptive decoder, query generation, consistency-based regularization. They show excellent improvement in recall@1 and recall@10 on the NQ320k datasets, largely outperforming previous baselines.

Weaknesses:

NCI is quite similar to DSI - e.g., the encoder-decoder architecture, indexing training and the hierarchical clustering algorithm to generate semantic doc IDs. It introduces several small improvements such as using prefixLM, consistency-based regularization, query generation, etc. The overall contribution seems limited.

Another weakness is that many of the gains come from using the additional query generation data. Interestingly, all other improvements seem to have very limited effect on the performance. This also raises a question about how much the model relies on the quality of the query generation model and how it can adapt to new domains that the QGEN model can’t generate good questions for the target document.

---

> ### Author Response · Authors · 2022-08-02
> **Addressing concerns**
>
> Thank you for reading the paper carefully and providing helpful feedback.
>
> We clarify that NCI achieves a significant improvement over DSI and SEAL even without query generation (53.63 for NCI v.s. 27.40/26.55 for DSI/SEAL on Recall@1). This improvement owes to the novel techniques we propose in this paper, including prefix-aware weight-adaptive (PAWA) decoder and consistency-based regularization.
>
> The ablation results without query generation are discussed in the general response section. Although query generation is important to the final performance, other techniques are also indispensable for the large improvement. In particular, replacing the PAWA decoder by a vanilla transformer decoder leads to a large performance drop (from 53.63 to 48.98 on Recall@1) in a fair comparison setting without query augmentation. The reason is that a vanilla transformer only utilizes fixed weight for next node prediction without considering its specific prefix in the tree. The PAWA decoder makes the routing weights in transformer adaptive to prefixes, and thus better leverages the semantic information incorporated in the hierarchical tree structure to improve the effectiveness of semantic document retrieval.

---

### Official Review · Reviewer_ncpT · 2022-07-13

**Rating:** 7
**Confidence:** 3
**Soundness:** 3 good
**Presentation:** 3 good
**Contribution:** 2 fair

**Summary:**

The paper proposed a sequence-to-sequence model that generates the relevant document given an input query, as a retrieval method. The proposed method have a huge improvement on the NQ320k dataset (Natural Questions). The most improvement seems to be coming from using generated query (based on the document) paired with the document as augmented training data.

**Questions:**

1. Can we use the same k-means procedure to obtain an identifier for a new document? I guess new documents will have the same identifier to be meaningful to the model?
2. It seems that the query generations has most contribution to the improvement. Have you compared with other query generation models / techniques so that we can understand what contribute to the success of this method?

**Limitations:**

The authors acknowledge a set of limitations of the model, including scale of the index set, inference speed and updating the index. I'm most interested in the third one. The model can only work on a fixed set of documents (to retrieve from), or there is a cost to update the model when the index set changes. This is a limitation of the line of works that generating the document identifiers, not specific to this work. However, I'm also curious, given that the document identifier is generated using a k-means algorithm, how the model performs when there are new documents if we just assign an identifier to a new document using the same k-means algorithm.

**Strengths And Weaknesses:**

Originality:
The proposed data augmentation technique is interesting and demonstrate good performance. The other techniques are interesting and providing much smaller but also good gains.

Quality:
The author addressed their limitations and present the work in context. The experiments are well designed.

Clarity:
The paper is clearly written.

Significance:
Very good gains on NQ320k over previous best method.

---

> ### Author Response · Authors · 2022-08-02
> **Thank you very much for the positive feedback.**
>
> We address your questions as follows.
>
> 1.Can we use the same k-means procedure to obtain an identifier for a new document? How does the model perform when there are new documents if we just assign an identifier to a new document using the same k-means algorithm? (Combining question 1 and questions in the Limitation part.)
> * In this paper, we focus on the problem where all documents are given at the outset. Supporting new documents requires augmentations to the proposed framework. For example, one approach is to assume that each leaf node in the tree is a semantic cluster, and each new document is added to one of the semantic clusters. In the retrieval stage, all documents under the same semantic cluster are retrieved if the corresponding cluster ID is generated by NCI. This approach is effective if a small fraction of documents are added to the corpus and there is no obvious topic drift. We are working in this direction and plan to report more empirical studies in the next version.
>
> 2.Have you compared other query generation models / techniques so that we can understand what contributes to the success of this method?
> * We leverage DocT5Query [1], which is a commonly used query augmentation method for document retrieval. As shown in Table 1, NCI outperforms BM25 + DocT5Query with a large margin (88.72 v.s. 58.39 on Recall@1), which shows that the NCI model is much more superior to BM25 based on the same query augmentation technique. Regarding the comparison without DocT5Query, the result is shown in the general response section. This verifies that NCI is the current best SOTA among all generative retrieval models even in the absence of query generation. Moreover, the ablation study shows that other proposed techniques, including the PAWA decoder, consistency-based regularization and semantic ID, also make indispensable contributions.
>
> [1] Rodrigo Nogueira, Jimmy Lin, and AI Epistemic. From doc2query to doctttttquery. Online preprint, 2019, https://github.com/castorini/docTTTTTquery.

---

### Official Review · Reviewer_V4yZ · 2022-07-18

**Rating:** 7
**Confidence:** 4
**Soundness:** 3 good
**Presentation:** 3 good
**Contribution:** 4 excellent

**Summary:**

While the traditional document retrieval problem is formulated as mapping both the document and the query to the same vector space and then performing nearest neighbor search to find the closest document fore each query, the paper instead proposes to formulate the task as a document identifier generation problem, where the input is the query and the output is the target document's id. This is not entirely novel with respect to DSI (Tay et al., 2022) but given that DSI was available on the archive only three months before NeurIPS deadline, one might be able to consider this paper to be a concurrent work. The proposed model by this paper, Neural Corpus Indexer, also introduces a few techniques: (1) query generation that augments the query-to-doc data for training (equivalent to document indexing), (2) prefix-aware weight-adaptive decoder that gives different symbols to the same tokens in different positions of the ids, (3) semantic document id so that similar documents share similar ids, and (4) regularization technique to prevent overfitting. The results on Natural Questions 320k (a subset of NQ) are very promising; the proposed model not only outperforms other generative retrieval models such as DSI and SEAL by a big margin but also strongly outperforms competitive baselines such as ANCE and BM25.

**Questions:**

- With regard to the weakness mentioned above, I am wondering how sensitive the model's performance is with respect to the quality of the generated queries.
- I don't think it is correct to say that "DSI hardly captures the hierarchical semantics of document identifiers" in L42. DSI indeed proposes a similar clustering mechanism to what this paper proposes.
- How is the documents clustered for semantic identifiers? Could you please give more description of Section 3.1.
- What is difference between p1 and p in L223? I think Section 3.4 is unclear.

**Limitations:**

Yes, the paper mentions crucial limitations such as its lack of scalability compared to canonical NNS retrieval system, which I agree as well.

**Strengths And Weaknesses:**

Strengths:
- The results are very strong. The baselines that the paper compares against, such as ANCE and BM25, are very strong ones and hard to beat. It is remarkable that a generative model can beat those baselines.
- Query generation helps a lot and this is a very useful information for people working in this field. In fact, as shown in Table 2, among other techniques introduced by the paper, QG is by far the biggest factor for the superior performance of NCI.

Weaknesses:
- If query generation is the dominant factor, then it seems to me that what the model is really doing is "query indexing" rather than "document indexing". This means the performance of the model will highly depend on generating a comprehensive range of queries which are likely to overlap with the test queries.
- Constructing the semantic hierarchy of the document ids can be considered as constructing the inverted index in the canonical NNS retrieval setup. Then what the decoder is doing can be considered as locating the correct cluster in each level of the hierarchy. Then I am wondering if it is fair to say this model is really "generative retrieval" or it is more of using different embeddings for the nearest neighbor search in each layer of the hierarchy.
- The paper only experiments on Natural Questions and only a subset of it. While NQ is a very representative dataset in this task, the paper would have looked much better if it was tested on different datasets and/or the full dataset of NQ. It is not clear why such results are not shown.

---

> ### Author Response · Authors · 2022-08-02
> **Thank you very much for the recognition of our work.**
>
> First, we address the three points you mentioned in the weaknesses part.
>
> 1. The goal of query generation is to generate training queries that share the same distributions with test queries and are semantically similar to the source documents. The generated queries do not necessarily overlap with the test queries in words. We calculate the overlapped ratio and only 12.45% of the test queries have appeared in the augmented training queries.
> 2. Our NCI model falls into the generative retrieval category as it is based on a sequence-to-sequence architecture and can be optimized end-to-end. The main difference between NCI and ANN algorithms using different embeddings is that the index structure of ANN is not differentiable while the entire NCI architecture is differentiable.
> 3. We follow the experimental setting of DSI, which evaluates on the NQ320k dataset. Experiments on the full set of NQ or other datasets require training larger models with more resources and time. We will report the results of more datasets in our next version.
>
> We answer the questions in the following.
>
> 1.How sensitive is the model’s performance with respect to the quality of the generated queries?
> * The quality of generated queries does matter a lot for the model’s performance. For instance, upgrading the query augmentation method from Doc2Query [1] to DocT5Query [2] improves the MRR@10 dev score of MSMARCO from 21.8 to 27.7 using BM25 backbone. However, we'd like to highlight that without DocT5Query, NCI still achieves noticeable improvements over state-of-the-art generative retrieval models. This shows that the novel architecture designs (e.g., the PAWA decoder) contribute significantly to the superior performance of NCI (see the result table in the general response section).  In this paper, we leverage DocT5Query as it is the most common query augmentation method used for document retrieval. We appreciate the suggestion to analyze the sensitivity of model performance with respect to the quality of generated queries. We will follow-up this in our future work.
>
> 2.I don't think it is correct to say that "DSI hardly captures the hierarchical semantics of document identifiers" in L42.
>
> * Yes, DSI also builds semantic identifiers through hierarchical K-means, but it uses a vanilla transformer decoder, which does not fully leverage the structural information. In NCI, we propose prefix-aware weight-adaptive (PAWA) decoder and consistency-based regularization loss to further guide the decoder to incorporate document semantics more effectively. In the PAWA decoder, a prefix-aware adaptive module is applied to the classification weights for token prediction, thus the tree-based decoder better leverages the semantic prior incorporated in the hierarchical tree structure. With consistency-based regularization loss, semantically similar queries will result in the same routing paths in the tree. Thank you for raising this question. We have polished the corresponding part in the revised version.
>
> 3.How are the documents clustered for semantic identifiers? Could you please give more description of Section 3.1.
>
> * We describe the full algorithm of document clustering in Algorithm 1 (Appendix B.2). The appendix PDF can be found in the zip file of supplementary material. In the revised version, we have added more details in Section 3.1 to make the content more self-contained. Specifically, each document is associated with one leaf node with a deterministic routing path $l=\{r_0, r_1, ..., r_m\}$ from the root, where $r_i \in [0, k)$ represents the internal cluster index for level $i$, and $r_m \in [0, c)$ is the leaf node. The semantic identifier for a document is concatenated by the node indices along the path from root to its corresponding leaf node. For documents with similar semantics, the prefixes of their corresponding identifiers are likely to be the same.
>
> 4.What is the difference between p1 and p in L223?
>
> * Thanks for pointing this out. This is a typo. It should be p1 and p2 for two independent query dropouts. Thus, we can add a consistency-based regularization loss in-between, which ensures that two augmented queries generate similar document identifiers.  We have polished Section 3.4 in the revised version.
>
>
> [1] Nogueira, Rodrigo, et al. "Document expansion by query prediction." arXiv preprint arXiv:1904.08375 (2019).
> [2] Rodrigo Nogueira, Jimmy Lin, and AI Epistemic. From doc2query to doctttttquery. Online preprint, 2019

---

> > ### Comment · Reviewer_V4yZ · 2022-08-08
> > **Thanks for the answers**
> >
> > Thanks for the answers. Most of my questions are answered and I support for the acceptance of the paper.

---

### Author Response · Authors · 2022-08-02
**General Response**

Dear reviewers,

Thank you for taking time in reading our paper and providing valuable comments. We briefly address common questions here.

1. Regarding the concern that query generation dominates final performance, we compare the performance of NCI and DSI/SEAL in a fair setting without query generation. The table below shows that NCI still outperforms the other two generative retrieval SOTAs significantly. Also, we provide ablations on the NCI version without query generation. The result shows that the PAWA decoder contributes a large improvement based on the raw transformer decoder (from 48.98 to 53.63 on Recall@1). It demonstrates that the PAWA decoder design is critically important in the absence of query generation. Semantic identifiers and consistency-based regularization are also beneficial.

    |                       | Recall@1 | Recall@10 |
    |-----------------------|----------|-----------|
    | DSI (T5-Base)         | 27.40    | 56.60     |
    | SEAL (BART-Base)      | 26.55    | 53.61     |
    | **NCI (T5-Base w/o QG)**  | **53.63**    | **67.84**     |
    | - w/o PAWA decoder    | 48.98    | --        |
    | - w/o Semantic id     | 50.78    | --        |
    | - w/o Consistency reg | 51.57    | --        |


2. Another contribution of this work is reproducibility. DSI does not provide open-source code, so we only refer to the numbers reported in its paper. We have released the code, data and checkpoints of NCI at an anonymous repository https://github.com/anonymousML36061/NCI. We believe the release will facilitate researchers to work on further comparisons and improvements of deep generative models for semantic text retrieval.

3. For easy reference of reviewers and ACs, all changes are highlighted in blue in the revision.

---

> ### Public Comment · Authors · 2022-12-05
> **Update**
>
> Note that the setting "NCI w/o QG" here corresponds to "NCI w/o DocT5Query" in the camera-ready paper. We use another kind of query augmentation, i.e., "document content as query", which is also adopted by DSI and SEAL. In the previous version, we use "doc as query" as a default setting and do not count it into query generation, while in the camera-ready version, we count it as one kind of query generation and perform more ablations. Moreover, we improve the consistency-based regularization from r-drop to contrastive loss and tune some hyper-parameters. Finally, the recall@1 of "NCI w/o DocT5Query" setting here has been improved from 53.63 to 60.23 (see Table 3 in the camera-ready version).

---

> ### Public Comment · ~Mu_Li3 · 2022-12-14
> **Performance difference between DSI and a bare bone NCI**
>
> Thanks for the ablation study that helps a lot to understand the effectiveness for various pieces. One question I have is that, by removing query generation, PAWA decoder and consistency reg, NCI looks quite similar to DSI (here I assume DSI also uses semantic identifiers based on DSI paper algorithm 1).  But the performance gap between this bare bone NCI and DSI is still huge. Would you mind explain the algorithm difference to help me understand?

---

> > ### Public Comment · Authors · 2022-12-14
> > **Thanks for your question.**
> >
> > One major difference between DSI and a bare bone NCI lies in the decoder vocabulary. When implementing the NCI decoder, we found that a shared vocabulary in different decoding steps cause huge performance drop, so we took each <position, id> pair as a unique token. For DSI, as not specified in the paper, we assume it still takes a standard transformer decoder with shared vocabulary + position embeddings. We are trying to figure out other differences and will update more ablations later in the arXiv version.

---

### Public Comment · Authors · 2022-12-04
**The revision comments**


We update the result tables in the camera-ready version. The revision is due to a different data version of query augmentation.
Previously, the data is cooked by one of our co-authors while using a different train-test split to train the query generator, causing some data leakage issue. All experiments in the previous submission are based on this query augmentation version, so the performance is relatively higher.  When preparing the camera-ready version, we review and reproduce the code end-to-end for official release.
At that time, we realize the data leakage problem. So, we re-cook the query augmentation data and reproduce all the experiments again in the new table. After solving the data leakage problem, NCI still shows more than 15% improvement over the current best SOTA.
We have released the complete open-source code at GitHub:

https://github.com/solidsea98/Neural-Corpus-Indexer-NCI

Welcome to follow and reproduce our work. Looking forward to further discussions and collaborations.

---

### Meta-Review · Area_Chair_JY8Q · 2022-08-22

**Recommendation:** Accept
**Confidence:** Certain

**Metareview:**

This paper  proposes a new framework for neural IR: given query, directly predict a document ID. The document IDs are obtained by hierarchical clustering of documents beforehand. This is a novel formulation of the problem, and is very distinct from current two-stage methods that have a high-recall sparse retrieval stage, followed by a high-precision neural reranker, or approximate nearest neighbor methods that encode both documents and queries as vectors. The paper is fairly well-written, the authors have addressed reviewers concerns with honest detailed feedback, and have made their code available to facilitate experimentation. The results are particularly strong compared to more traditional BM25-based models and competing neural approaches like DSI. I anticipate there will be much follow-on work and eventually a paradigm shift in neural IR.

**Award:**

Yes

---

### Decision · Program_Chairs · 2022-09-14

Accept